# Mixture of Inputs:
# Text Generation Beyond Discrete Token Sampling

**Yufan Zhuang**[1], **Liyuan Liu**[2], **Chandan Singh**[2], **Jingbo Shang**[1], and **Jianfeng Gao**[2]

[1]UC San Diego
[2]Microsoft Research

## Abstract

In standard autoregressive generation, an LLM predicts the next-token distribution, samples a discrete token, and then discards the distribution, passing only the sampled token as new input. To preserve this distribution's rich information, we propose Mixture of Inputs (MoI), a training-free method for autoregressive generation. After generating a token following the standard paradigm, we construct a new input that blends the generated discrete token with the previously discarded token distribution. Specifically, we employ a Bayesian estimation method that treats the token distribution as the prior, the sampled token as the observation, and replaces the conventional one-hot vector with the continuous posterior expectation as the new model input. MoI allows the model to maintain a richer internal representation throughout the generation process, resulting in improved text quality and reasoning capabilities. On mathematical reasoning, code generation, and PhD-level QA tasks, MoI consistently improves performance across multiple models including QwQ-32B, Nemotron-Super-49B, Gemma-3-27B, and DAPO-Qwen-32B, with no additional training and negligible computational overhead.

## 1 Introduction

Large language models (LLMs) are trained to predict the full distribution of the next token given an input context. To generate desirable sequences of text, various methods have been proposed to sample discrete tokens from these iterative next-token distributions [1, 2]. After the sampling process, only the discrete token is passed as the new input, and the rich predicted distribution is discarded. This process forces the model to commit to a single path in its reasoning, potentially abandoning valuable alternatives that could lead to better solutions.

On the other hand, human thinking first occurs in a high-dimensional and fluid manner before being articulated as natural language. Inspired by this cognitive process, we explore methods to enable LLMs to utilize not only articulated natural language but also partially-formed ideas, competing possibilities, and conceptual associations that exist in a probabilistic space before crystallizing into words.

Specifically, we propose Mixture of Inputs (MoI), a novel approach that takes as input not only a discrete, sampled token but also the sampled token's distribution. This preserves the model's uncertainty and allows it to conduct inner speech in a high-dimensional space. We employ a Bayesian estimation method, treating the token distribution as the prior and the sampled token as the observation, then replacing the conventional one-hot vector with the continuous posterior expectation. With this posterior expectation, a weighted average embedding is passed as the new input to subsequent prediction steps.

---

Code is available at: `https://github.com/EvanZhuang/mixinputs`.

39th Conference on Neural Information Processing Systems (NeurIPS 2025).

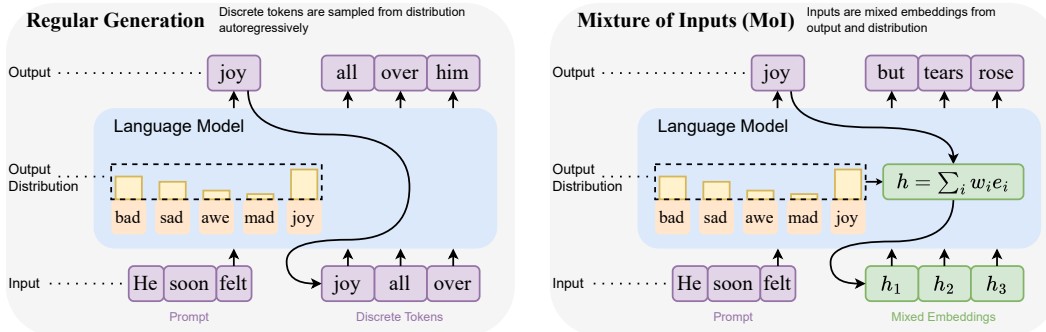

Figure 1: Comparison of the regular autoregressive generation pipeline (left) and our proposed Mixture of Inputs (MoI) strategy (right). In regular generation, only the discrete sampled token is fed back at each step, whereas MoI preserves the full sampling distribution by computing a blended embedding $h = \sum_i w_i e_i$, with weights $w_i$ interpolating embeddings $\{e_i\}_{i=1}^{V}$, letting the model consider several plausible tokens simultaneously within a single forward pass.

MoI is conceptually intuitive and requires no additional training or architectural changes, making it immediately applicable to existing models. We implemented our method in modern LLM serving frameworks and found it to have negligible computational overhead and minimal deployment effort.

We evaluate MoI across a range of tasks—including mathematical reasoning, code generation, and graduate-level question answering—where maintaining uncertainty can play a crucial role in step-by-step inference. Across these domains, MoI brings consistent performance improvements for multiple models including QwQ-32B, Nemotron-Super-49B, Gemma-3-27B, and DAPO-Qwen-32B.

## 2    Related Work

**Linearity of Embedding Representations**    The foundation of our work builds upon emerging research on the continuous nature of language model embedding spaces. Semantic linearity has been observed in embedding spaces dating back to word embedding models [3] and has been shown in various ways in modern LLMs [4–7]. A more recent work demonstrates that transformer language models naturally learn to process inputs and outputs as smooth, continuous functions rather than merely as discrete tokens [8]. This finding suggests that models inherently operate in a continuous latent space, even when traditionally constrained to discrete token processing. Similarly, Vector-ICL [9] shows that LLMs can effectively decode and process projected text embeddings via linear projection layers when provided with in-context demonstrations. While Vector-ICL projects external continuous data into the embedding space, our MoI directly leverages the linearity of the existing embedding space, demonstrating that meaningful representations can be created through linear combinations of token embeddings. Our work extends these insights by applying them specifically to preserve distributional information during the generation process, showing that this approach can enhance reasoning capabilities without model modifications.

**Continuous Chain of Thought**    Chain-of-thought (CoT) prompting and related works improve language model performance by encouraging step-by-step reasoning through natural language [10–12]. However, these approaches rely on discrete text tokens, which can become inefficient and lengthy. More recently, COCONUT (Chain of Continuous Thought) [13] addresses this limitation by operating directly in the model's hidden state space rather than generating explicit text. By feeding the model's hidden state back as input, COCONUT enables more efficient reasoning that condenses lengthy thoughts into single tokens without the overhead of explicit thought generation. While COCONUT manipulates hidden states during multi-step reasoning processes, our MoI similarly leverages continuous representations but focuses specifically on the input embedding space during token generation. This key difference allows our approach to achieve improved reasoning without requiring architectural changes or model retraining, making it a more lightweight and accessible intervention.

**Prompt and Weight Merging**   Linearity of LLM representations has been explored in a few related applications. Motivated by the success of methods that improve performance by ensembling multiple LLM calls [14–16], learning an ensemble of soft prompts or compressing a large prompt have been studied to enable strong performance without increasing computational cost [17–19]. Similarly, mechanistic methods for steering have proposed adding different latent vectors to elicit desired LLM behaviors [20–22]. The concept of linearity in neural networks extends beyond input representations to model parameters themselves. Recent work demonstrates that when two language models with shared initialization are combined through linear interpolation of their weights, their internal representations blend to produce a stronger model [23]. This discovery has enabled various model-merging techniques, from basic weight averaging to more sophisticated approaches [24, 25]. MOI applies similar linearity principles but at the level of individual tokens rather than full prompts or model weights.

## 3   Methods: Mixture of Inputs

When humans think, they often use natural language as an internal dialogue, but thinking is more fluid and multidimensional than just discrete words and sentences. Our cognition includes partially-formed ideas, competing possibilities, and conceptual associations that exist in a probabilistic space before crystallizing into specific language.

Our proposed method mirrors this cognitive reality by enabling LLMs to take as inputs both discrete tokens (representing specific linguistic choices) and token distributions (capturing the uncertainty, nuance, and competing possibilities that exist in human thought). By combining both as the model input, we obtain a richer representation that better reflects how human thinking operates — balancing the concrete and the probabilistic aspects of cognition.

Specifically, we introduce Mixture of Inputs (MOI). The core idea is to reinterpret token mixing as probabilistic inference under a Bayesian model. This formulation enables a principled mechanism to reconcile the model's prior belief (the output distribution) with its observed evidence (the sampled tokens), resulting in a more robust and statistically grounded method for input blending.

### 3.1   Token Generation and Embedding Aggregation

A key strength of MOI lies in its simplicity and modularity: it enhances the input representation without altering the model architecture or the underlying sampling algorithm. MOI operates after the language model produces its output distribution and before the next token is fed back into the model for the subsequent generation.

**Token Generation**   Let $\{e_i\}_{i=1}^{V} \in \mathbb{R}^d$ be embedding weights, with hidden dimension $d$ and vocabulary size $V$. At each decoding timestep $t$, the language model outputs a probability distribution $\boldsymbol{p}_t = \{p_{t,i}\}_{i=1}^{V}$ over the vocabulary. This is typically followed by a sampling step that selects a token $\boldsymbol{y}_t$ (e.g., via top-$k$, nucleus sampling, or temperature scaling). In conventional approaches, the model would retrieve the embedding $\boldsymbol{e}_{y_t}$ corresponding to the sampled token and feed it into the next layer as the sole input.

MOI does not modify the sampling process itself: the sampled token $y_t$ is still used as the output token. This design makes MOI fully compatible with any decoding strategy and seamlessly integrable into existing autoregressive generation pipelines.

**Embedding Aggregation**   MOI first uses both the sampled token $\boldsymbol{y}_t$ and the distribution $\{p_{t,i}\}$ to compute $\{w_{t,i}\}$ as in Section 3.2, then uses $\{w_{t,i}\}$ to construct a *mixed embedding* vector $\boldsymbol{h}_t$.

$$\boldsymbol{h}_t = \sum_{i=1}^{V} w_{t,i}\boldsymbol{e}_i, \quad \text{where} \quad w_{t,i} \geq 0, \ \sum_i w_{t,i} = 1. \tag{1}$$

This representation allows the model to reason over a distribution of plausible next tokens rather than committing to a single discrete choice, effectively enabling a form of "inner speech" with richer representational capacity.

### 3.2 Bayesian Input Construction with MOI

To capture the distribution information, a naive idea might simply be to directly mix the inputs according to the output distribution, setting $w_{t,i} = p_{t,i}$. However, this approach only treats the token distribution as the input and neglects the sampled next token. In Section 6.1, we experiment with this approach (referred to as *Direct Mixture*) and find that it leads to performance degradation in most cases.

Instead, MOI combines two sources of information: (1) the output distribution $p_t$, representing the model's prior belief over possible next tokens, and (2) the sampled token $y_t$, representing a concrete observation drawn from this belief.

To reconcile these two sources, MOI treats the sampling process as probabilistic evidence and formulates the blending of representations as a Bayesian inference problem. Specifically, it constructs a posterior-weighted mixture over token embeddings by computing a new weight vector $w_t = \{w_{t,i}\}_{i=1}^V$ that incorporates both the uncertainty in $p_t$ and the evidence from $y_t$.

The resulting mixed embedding $h_t$ is given by Equation 1, and it replaces the embedding for the discrete token as the input to the next decoding step (i.e., replaces $e_{y_t}$ with $h_t$). What changes is the internal representation passed into the model, allowing the decoder to reason over both the chosen token and the context of plausible alternatives.

## 4 Mixing Weight Estimation

Here, we elaborate our proposed Bayesian estimation method for $w_t = \{w_{t,i}\}_{i=1}^V$.

### 4.1 Dirichlet Mixture Model

In probabilistic modeling, a prior encodes belief before observing new data. Accordingly, we begin by constructing a prior distribution over token choices based on the model's output logits. Specifically, we assume the prior distribution to be Dirichlet, with concentration parameter $\alpha$.

We view $y$ as the output of the sampling process and assume the sampled token comes from a multinomial distribution parametrized by $w$. Then, we estimate the mixing weight $w$ by conducting the posterior estimation.

$$
\begin{aligned}
&w \sim \mathrm{Dir}(\alpha), && \text{where } \alpha = \mathrm{H}(p) \cdot p, \\
&y \sim \mathrm{Multinomial}(w) && \mathrm{H}(p) \text{ is the normalized entropy of } p
\end{aligned}
$$

This formulation ensures that tokens with higher model confidence (i.e., lower entropy) exert stronger influence on the posterior, while still respecting the sampled outcome. We will go over each part of the Bayesian model in the following sections.

### 4.2 Estimating Mixing Weight

Let $p_t \in \Delta^{V-1}$ be the next-token distribution at step $t$ and let $y_{t,i} \in \{0,1\}$ indicate the sampled token ($y_{t,i} = 1$ iff token $i$ is chosen). We estimate the mixing weights $w_t$ by Bayesian posterior inference in a Dirichlet–Multinomial model.

**Entropy-scaled prior.** Define the *normalized entropy $H$* as the following

$$
H := \mathrm{H}(p_t) = -\frac{1}{\log V} \sum_{i=1}^{V} p_{t,i} \log p_{t,i} \qquad H \in [0,1]. \tag{2}
$$

We place a Dirichlet prior

$$
w_t \sim \mathrm{Dir}(\alpha), \qquad \alpha = \mathrm{H}(p_t)\, p_t, \tag{3}
$$

so that the total concentration $\sum_i \alpha_i = \mathrm{H}(p_t)$ grows with uncertainty and vanishes when the model is confident. So when uncertainty is high, the prior distribution will be more widespread over $p_t$, and vice versa.

**Pseudo-count observation.** The sampled token contributes a single pseudo-count whose weight increases as confidence rises:

$$c_i = \big(\beta + 1 - H\big)\, y_{t,i}, \qquad \text{with hyperparameter } \beta. \tag{4}$$

The hyperparameter $\beta$ controls the concentration over mixing weight. Smaller values emphasize more on output distributions, while larger values highlight more the sampled output token. The effect of $\beta$ is easier to observe in Eq. (5). We also conduct an analysis of $\beta$'s empirical effect in Section 7.1.

**Posterior mean.** Dirichlet conjugacy yields the posterior mean of $\boldsymbol{w}_t$, and we use that estimation as our mixing weights:

$$w_{t,i} = \frac{\boldsymbol{\alpha}_i + c_i}{\sum_i \boldsymbol{\alpha}_i + N} = \frac{H\, p_{t,i} \;+\; \big(\beta + 1 - H\big)\, y_{t,i}}{\beta + 1}, \quad \text{with } N = \sum_i c_i \tag{5}$$

**Behavior of $w$.** Eq. (5) smoothly interpolates between the distribution ($w_{t,i} \to p_{t,i}$ when $H \to 1$) and the one-hot token ($w_{t,i} \to y_{t,i}$ when $H \to 0$), thereby reconciling distributional and discrete evidence in a single principled estimator.

The complete procedure for computing the mixture of inputs is summarized in Algorithm 1.

## 5 Experimental Setup

We evaluate MOI across a diverse suite of benchmarks spanning competition mathematics, combinatorial problem solving, program synthesis, and graduate-level question answering. These tasks vary widely in structure and domain, allowing us to assess MOI's generality and effectiveness across distinct application settings.

---

**Algorithm 1:** Mixture of Inputs

**Require:** Sampling distribution $\boldsymbol{p}_t$, sampled token $\boldsymbol{y}_t$, hyperparameter $\beta$, and embeddings $\{e_i\}_{i=1}^V$.
1. Compute entropy $H$ with Eq. 2
2. Compute mixing weight $w_{t,i}$ with Eq. 5
**return** $\boldsymbol{h}_t = \sum_i w_{t,i}\, \boldsymbol{e}_i$

---

### 5.1 Tasks and Metrics

To ensure a comprehensive evaluation, we select four challenging benchmarks that span distinct reasoning domains and require different cognitive skills, from symbolic manipulation to procedural generation and scientific comprehension:

**AIME** [26] consists of complex high-school level mathematical problems that often require multiple stages of symbolic reasoning, algebraic manipulation, and geometric insight. We use the official AIME datasets from 2022 to 2024 and evaluate models based on exact match accuracy, reflecting their ability to arrive at precise, correct solutions.
**Count Down 4** [27] is a synthetic numerical reasoning task that presents models with arithmetic puzzles. It requires deriving a target number by applying a sequence of operations (addition, subtraction, multiplication, division) on a fixed set of four input numbers. This benchmark emphasizes procedural and combinatorial reasoning. We report the success rate, indicating whether the model arrives at the correct final equation.
**LiveCodeBench** [28] is a dynamic and realistic code generation benchmark that includes tasks ranging from simple string manipulations to advanced data structures and algorithms. Each problem specifies a goal in natural language, and the model must generate executable code that meets functional correctness criteria. We use pass@1—the proportion of correct solutions on the first attempt—as the primary evaluation metric.
**GPQA** [29] is a highly challenging multiple-choice question answering benchmark drawn from graduate-level science and engineering exams. Its diamond subset features the most difficult questions that demand domain-specific knowledge, long-range reasoning, and the integration of multiple concepts. We evaluate models based on multiple-choice accuracy.

### 5.2 Models

We evaluate MOI using 4 state-of-the-art open-source LLMs with advanced reasoning capabilities.

**QwQ-32B** [30] is optimized for mathematical and logical reasoning through a curriculum of instruction tuning on symbolic tasks, math word problems, and chain-of-thought datasets.

**Llama-3.3-Nemotron-49B** [31] is derived from Meta's Llama 3.3 70B model [32]. The model underwent neural architecture search to optimize for inference efficiency, followed by supervised fine-tuning and reinforcement learning. These techniques were applied to enhance the model's reasoning abilities, instruction following capabilities, and tool-calling performance.

**Gemma-3-27B** [33] is part of Google's Gemma 3 family—multimodal (text + image) models with 128 K token context windows and an integrated SigLIP vision encoder. The 27B variant is instruction-tuned for chat and reasoning.

**DAPO-Qwen-32B** [34] is a customized version of Qwen2.5-32B [35] that incorporates Decoupled Clip and Dynamic Sampling Policy Optimization (DAPO), which stabilizes and scales RL for long chain-of-thought reasoning. This model is designed to encourage faithful and step-consistent reasoning trajectories.

### 5.3 Baselines

To quantify the benefit of MOI, we compare it with two decoding schemes that keep the underlying model architecture and sampling mechanism fixed. The primary baseline (*Standard*) is the widely used nucleus sampling with temperature scaling [1]. It represents the default inference recipe shipped with each model. Our second baseline (*Direct Mixture*) constructs the input representation as a simple weighted sum of token embeddings using the softmax probabilities as coefficients, i.e., computing the value of $h_t$ as $\sum_{i=1}^{V} p_{t,i} e_i$. Unlike MOI, it performs no Bayesian reconciliation between the distribution and the sampled token, providing a stringent ablation for assessing the value of our posterior estimator. We also tried directly feeding the mixed output hidden states, but we found that the models cannot make sense of the hidden states without retraining.

### 5.4 Hyperparameter Settings

We perform 5 runs for all experiments and report the average. For AIME and Count Down 4, we perform hyperparameter grid search on baselines, Direct Mixture and MOI with $\beta \in \{\frac{1}{4}, \frac{1}{2}, 1, 2, 4, 8\}$, $T \in \{0.6, 0.8, 1\}$ and top-p $\in \{0.4, 0.6, 0.8, 0.95\}$. We report the mean result of the best configuration for all three methods. We investigate the importance of these hyperparameters in Section 6.2. For GPQA-Diamond and LiveCodeBench, we use the universal hyperparameter for all of them with $T = 0.6$, top-p $= 0.95, \beta = 1$; more details can be found in Appendix F.

## 6 Main Results

### 6.1 MOI Boosts Capabilities of LLMs

Table 1 reports accuracy on four reasoning-intensive benchmarks for four open-source LLMs. Across all 16 model–task pairs, our approach MOI either matches or outperforms the *Standard* autoregressive baseline, with an average absolute gain of 1.8%. In contrast, the ablation that removes distribution–smoothing (*Direct Mixture*) degrades performance in most cases, underscoring the importance of our Bayesian smoothing.

**Consistency across model scales.** MOI achieves gains for both medium-sized (Gemma-3-27B) and larger (32 to 49 B-parameter) models. The largest improvement appears on Nemotron-Super-49B, where MOI adds up to +4.1% on GPQA-Diamond and +2.6% on Count Down 4, lifting the overall average to 55.45% (+2.36%). These results indicate that mixture-of-inputs remains beneficial even when the underlying model already possesses strong zero-shot reasoning abilities.

**Task-specific trends.** Improvements are most pronounced on benchmarks requiring extended symbolic manipulation. Count Down 4 benefits the most (+3.7% mean gain), suggesting that explicitly representing uncertainty over arithmetic operations mitigates the compounding error typical in multi-step numerical reasoning. Gains on AIME and GPQA-Diamond further show that MOI generalizes from high-school mathematics to graduate-level science QA, while LiveCodeBench sees more modest but still positive changes.

Table 1: Main results on four benchmarks with four large language models. The "Input Info." column indicates the source of input passed into the model: *Output Token* uses only the sampled discrete token, *Output Dist.* uses the full output probability distribution, and *Token + Dist.* combines both. Accuracy (%) is reported on AIME, Count Down 4, GPQA-Diamond, and pass@1 is used on LiveCodeBench. *Standard* uses conventional sampling, that is temperature–scaled nucleus sampling, *Direct Mixture* removes the posterior estimation, and MOI is our full approach. Shaded cells highlight MOI and its performance gain (absolute difference over the conventional generation).

| Model | Method | Input Info. | AIME | CountDown4 | GPQA-D | LiveCodeBench | Avg |
|---|---|---|---|---|---|---|---|
| **QwQ-32B** | Standard | Output Token | 77.78 | 79.25 | 58.08 | 76.32 | 72.86 |
| | Direct Mixture | Output Dist. | 72.00 | 66.88 | 51.52 | 53.42 | 60.96 |
| | MOI | Token + Dist. | 80.00 | 80.01 | 60.10 | 76.51 | 74.15 |
| | Gain vs. Standard | | **+2.22** | **+0.76** | **+2.02** | **+0.19** | **+1.29** |
| **Nemotron-Super-49B** | Standard | Output Token | 54.89 | 56.93 | 60.60 | 39.92 | 53.09 |
| | Direct Mixture | Output Dist. | 60.00 | 51.72 | 60.10 | 16.04 | 46.97 |
| | MOI | Token + Dist. | 57.11 | 59.53 | 64.65 | 40.50 | 55.45 |
| | Gain vs. Standard | | **+2.22** | **+2.60** | **+4.05** | **+0.58** | **+2.36** |
| **Gemma-3-27B** | Standard | Output Token | 25.56 | 56.51 | 46.97 | 31.31 | 40.09 |
| | Direct Mixture | Output Dist. | 26.44 | 55.47 | 51.52 | 31.99 | 41.36 |
| | MOI | Token + Dist. | 26.89 | 59.38 | 47.47 | 32.87 | 41.65 |
| | Gain vs. Standard | | **+1.33** | **+2.87** | **+0.50** | **+1.56** | **+1.56** |
| **DAPO-Qwen-32B** | Standard | Output Token | 64.67 | 72.03 | 42.42 | 54.01 | 58.28 |
| | Direct Mixture | Output Dist. | 62.67 | 67.19 | 37.88 | 23.87 | 47.90 |
| | MOI | Token + Dist. | 64.44 | 78.75 | 42.93 | 55.18 | 60.33 |
| | Gain vs. Standard | | **-0.23** | **+6.72** | **+0.51** | **+1.17** | **+2.05** |

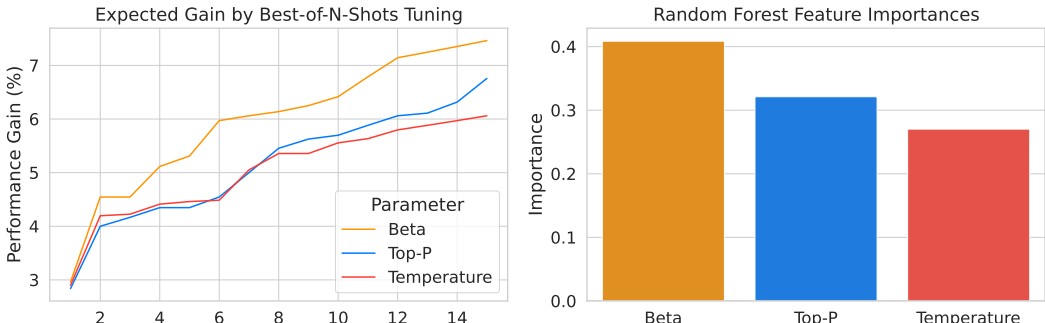

Figure 2: **Hyperparameter Importance Analysis.** Comparison of three key hyperparameters ($\beta$ in MOI, top-p, and temperature) across four LLMs on two mathematical reasoning tasks. **Left:** Expected performance gain (%) when optimizing each hyperparameter individually through best-of-N-shots tuning. The graph shows $\beta$ consistently outperforms other parameters as N increases. **Right:** Relative feature importance derived from random forest regression analysis, confirming $\beta$'s strong influence (0.41) on model performance compared to top-p (0.32) and temperature (0.27). These results demonstrate that $\beta$ is highly influential for effectively controlling input mixing during chain-of-thought reasoning.

**Role of autoregressive inputs.** Feeding back the *full* output distribution alone is insufficient: *Direct Mixture* often harms accuracy (e.g., -22.9% on LiveCodeBench for Nemotron-Super-49B). The combination of the sampled token *and* its distributional context lets the model retain a discrete anchor while preserving alternative hypotheses, yielding the best of both worlds.

Together, these findings demonstrate that MOI offers a principled and consistently effective way to enhance multi-step reasoning. By marrying discrete choices with probabilistic context, it improves accuracy without sacrificing decoding efficiency or requiring model-specific fine-tuning.

## 6.2 Hyperparameter Importance Analysis

To understand which factors most strongly influence reasoning performance, we analyze three key hyperparameters: $\beta$, top-p, and temperature. This analysis spans four LLMs and two mathematical

reasoning tasks, with multiple runs and grid search over the hyperparameter space, as described in Section 5.4.

Fig. 2 provides two complementary perspectives on hyperparameter importance. The left plot tracks expected performance gain when optimizing each parameter individually through best-of-N-shots tuning, with experiment setup explained in Appendix E.1. As N increases from 1 to 15, $\beta$ consistently yields the highest gains, reaching nearly 7.5% improvement at N=15, while top-p and temperature plateau at approximately 6.0-6.5%. This separation becomes particularly pronounced after N=10, suggesting that $\beta$'s impact grows with more extensive search.

The right panel quantifies each parameter's importance through random forest regression analysis, with experiment setup explained in Appendix E.2. With inputs as hyperparameters and accuracy as the target, this reveals $\beta$ as the dominant factor (importance score of 0.41), followed by top-p (0.32) and temperature (0.27).

## 7 Analysis

### 7.1 Task-Dependent Optimal Mixing Strategies

Different reasoning tasks may benefit from varied degrees of distribution mixing. To investigate this phenomenon, we analyze the parameter sensitivity of two distinct benchmark types: **AIME** (requiring advanced mathematical reasoning) and **Count Down 4** (demanding extensive combinatorial enumeration). Fig. 3 visualizes how performance varies with the mixing parameter $\beta$ across four LLMs, showing the deviation from each task's global mean accuracy.

The results reveal an interesting inverse relationship between task type and optimal $\beta$ values. AIME performance peaks at low $\beta$ values ($\beta \leq 1$), with accuracy dropping sharply when $\beta > 1$. In contrast, Count Down 4 shows the opposite pattern, performing substantially below average at low $\beta$ values but excelling when $\beta > 1$. This divergence suggests fundamental differences in how distribution mixing affects distinct reasoning processes.

For reasoning-intensive AIME problems, low $\beta$ values promote greater consideration of alternative solution paths while maintaining focus on the most promising directions. Conversely, for enumeration-intensive Count Down 4 problems, higher $\beta$ values increase concentration on the most probable combinations, effectively pruning the vast search space.

These findings highlight the importance of task-appropriate $\beta$ calibration when deploying MoI. Lower values suit open-ended reasoning, while higher values suit systematic enumeration—an adaptability that fixed decoding strategies lack.

### 7.2 Case Study: Linear Prompt Blending with Various Lengths

Although our main experiments focus on token-by-token blending at generation time, we also investigate whether a similar blending strategy applied to instruction prompts of varying lengths can boost performance. To this end, we perform 10-shot in-context learning on five sentiment analysis benchmarks using three medium-sized LLMs, building on prior work showing that prompt wording and structure have a major impact on classification accuracy [15].

We assembled three prompt pools: (1) binary sentiment analysis on Rotten Tomatoes [36], SST2 [37], and IMDB [38], consisting of 96 prompts of length 3–16 words (mean 7.57); (2) 6-class emotion classification on the Emotion dataset [39], with 32 prompts of length 3–15 words (mean 7.27); and (3) 3-class financial sentiment on the Financial Phrasebank [40], comprising 40 prompts of length 2–14 words (mean 6.51).

Our blending procedure first linearly extrapolates each prompt's embedding to the maximum length in its pool and then averages these fixed-length embeddings to form a single "blended" prompt representation. This approach integrates semantic nuances from all constituent prompts while preserving their instructional intent.

Table 2 reports 10-shot accuracy under the expectation over randomly drawn single-prompt, the blended-prompt, and the absolute gain over baseline. Across most benchmarks and models, linear prompt blending consistently outperforms random single-prompt selection, further demonstrating that embedding-space mixing can be highly effective for boosting LLMs' capacity.

Table 2: 10-shot in-context learning accuracy (%) for three LLMs (Llama3 8B [32], Mistral 7B [41], Gemma 7B [42]) on five sentiment analysis benchmarks. We compare the expectation of a single-prompt baseline against embedding-space prompt blending via linear interpolation of prompts.

| Model | Method | Rotten Tomatoes | SST2 | IMDB | Emotion | Financial Phrasebank | Average |
|---|---|---|---|---|---|---|---|
| Llama3 8B | Single Prompt | 91.58 | 94.12 | 87.26 | 53.82 | 68.76 | 79.11 |
| | Linear Interpolation | 92.68 | 94.49 | 95.40 | 51.75 | 72.34 | 81.33 |
| | Gain | **+1.10** | **+0.37** | **+8.14** | **-2.07** | **+3.58** | **+2.22** |
| Mistral 7B | Single Prompt | 89.32 | 91.11 | 85.06 | 54.87 | 70.75 | 78.22 |
| | Linear Interpolation | 92.21 | 94.03 | 92.82 | 51.60 | 73.42 | 80.82 |
| | Gain | **+2.89** | **+2.92** | **+7.76** | **-3.27** | **+2.67** | **+2.59** |
| Gemma 7B | Single Prompt | 86.66 | 87.18 | 87.31 | 50.77 | 72.63 | 76.91 |
| | Linear Interpolation | 92.30 | 93.34 | 93.88 | 50.30 | 74.39 | 80.84 |
| | Gain | **+5.64** | **+6.16** | **+6.57** | **-0.47** | **+1.76** | **+3.93** |

## 7.3 Throughput Analysis

The mixing weight calculation is lightweight and efficient. We perform a throughput analysis, shown in Table 3, to examine the runtime overhead added by MOI. We measure generation statistics for solving the Count Down 4 task, and we record the average input and output throughput over 5 runs. To better compare the throughput, we picked the benchmarks where the generation length is about the same. The median difference between MOI-generated text and baseline-generated text is 1.7%.

Table 3: Throughput analysis (tokens/s) for QwQ-32B with and without MOI in vLLM.

| Method | Input Speed | Output Speed |
|---|---|---|
| Standard | 62.87 | 1,143.31 |
| MOI | 61.36 | 1,101.44 |
| Overhead | 2.40% | 3.66% |

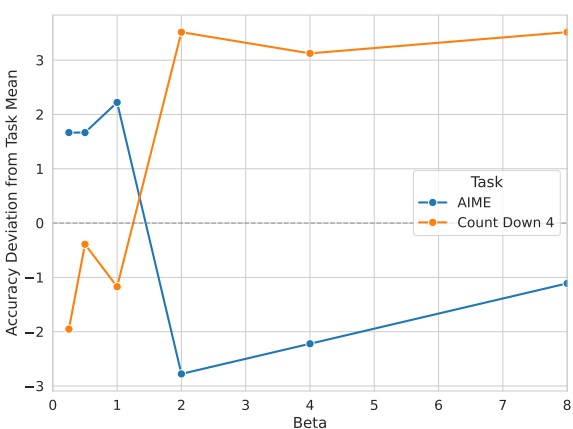

Figure 3: **Task-dependent Optimal Mixing Strategies.** The plot shows accuracy deviation from task mean across different $\beta$ values for AIME (reasoning-heavy) and Count Down 4 (enumeration-heavy), averaged across four LLMs. Lower $\beta$ values ($\beta \leq 1$) significantly benefit AIME's performance while higher $\beta$ values ($\beta > 1$) improve Count Down 4. This divergence demonstrates how MOI's impact varies based on task characteristics: reasoning-intensive tasks perform better with stronger distribution mixing (low $\beta$) to be more creative, while enumeration-intensive tasks benefit from higher distribution mixing (high $\beta$) that helps explore the combinatorial search space with more focus.

## 8 Discussion

**Limitations and Future Work** While MOI demonstrates consistent gains on a wide range of benchmarks, its current scope is intentionally focused on tasks that can be objectively evaluated. As a result, applications such as open-ended generation or creative writing, where objectives are less formally defined, remain outside the current scope and present promising directions for further study. Additionally, we observe that the hyperparameter $\beta$ exhibits task-dependent behavior. This suggests that different task types benefit from varying degrees of distributional mixing, a phenomenon worthy of deeper theoretical exploration. Future work could investigate adaptive or test-time $\beta$ tuning strategies.

**Conclusion** We presented Mixture of Inputs (MOI), a training-free enhancement to autoregressive generation that preserves distributional information. By treating input mixing as a Bayesian inference problem, MOI maintains a richer internal representation throughout the generation process, allowing models to conduct a form of inner speech beyond discrete tokens while requiring no architectural changes or additional training.

Our evaluation across LLMs and benchmarks demonstrates consistent performance improvements. MOI's conceptual simplicity, negligible computational overhead, and immediate applicability to existing models make it a practical enhancement that bridges the gap between the high-dimensional nature of thought and the discrete nature of language.

## Acknowledgement

Our work is sponsored in part by NSF CAREER Award 2239440, NSF Proto-OKN Award 2333790, Sponsored Research Projects from companies like Cisco and eBay, as well as generous gifts from Google, Adobe, and Teradata. Any opinions, findings, and conclusions or recommendations expressed herein are those of the authors and should not be interpreted as necessarily representing the views, either expressed or implied, of the U.S. Government. The U.S. Government is authorized to reproduce and distribute reprints for government purposes not withstanding any copyright annotation hereon.

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

# A    Comparing over Hyperparameter Grid Search

We perform a head-to-head evaluation between our method (MOI) and the standard text generation with temperature-scaled nucleus sampling (baseline), under two complementary regimes:

**Best-case:** each method is run with its single best-performing hyperparameter configuration, Fig. A1 summarizes the results.

**Grid-average:** performance is averaged across all combinations in the hyperparameter grid (see details in Section 5.4). Fig. A2 provides these averages and confirms that the gains are not an artifact of cherry-picking one lucky setting.

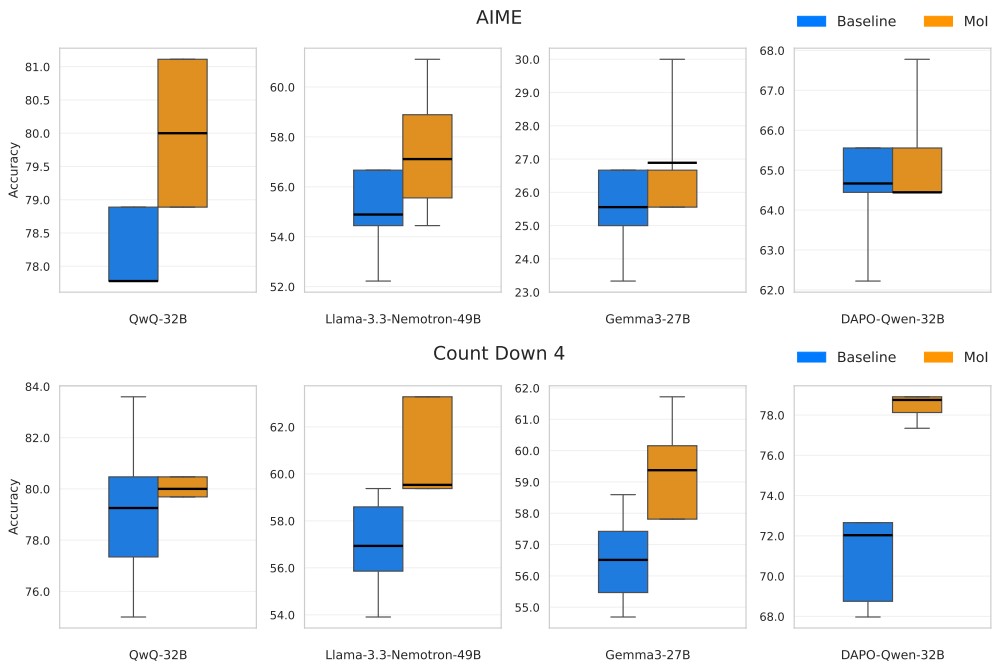

Figure A1: We show a comparison of distributions of evaluation results across the best top-$p$ and temperature hyperparameter for baseline and with MOI. The results indicate strong performance gain brought by incorporating the sampling distribution in the generation process.

# B    Generalization of a single hyper-parameter setting across tasks

A practical concern when adding new decoding hyperparameters is whether the values tuned on one task will transfer to others. To investigate this, we conduct a grid search over the hyperparameters described in Section 5.4. For each (method, model) pair, we keep the configuration that maximizes the average accuracy on AIME & CountDown 4 and freeze it for the remainder of the study.

Table A1 shows the results on the held-out GPQA-Diamond and LiveCodeBench. The single-tuned setting consistently outperforms Standard nucleus sampling on 15 of the 16 (model, task) pairs.

Overall, these results suggest that MOI requires only modest tuning effort: once the hyperparameters are calibrated on a small proxy set, they generalize robustly to new tasks and domains without much further adjustment.

# C    Results on Instruction-following Tasks

We conducted additional experiments on MT-Bench [43], using the four larger models, and the two smaller models, Llama-3.1-8B [32] and Qwen-2.5-14B [35]. Below, we report the average performance for standard decoding (temperature=0.6, top-p=0.95) and our Mixture of Inputs (MOI)

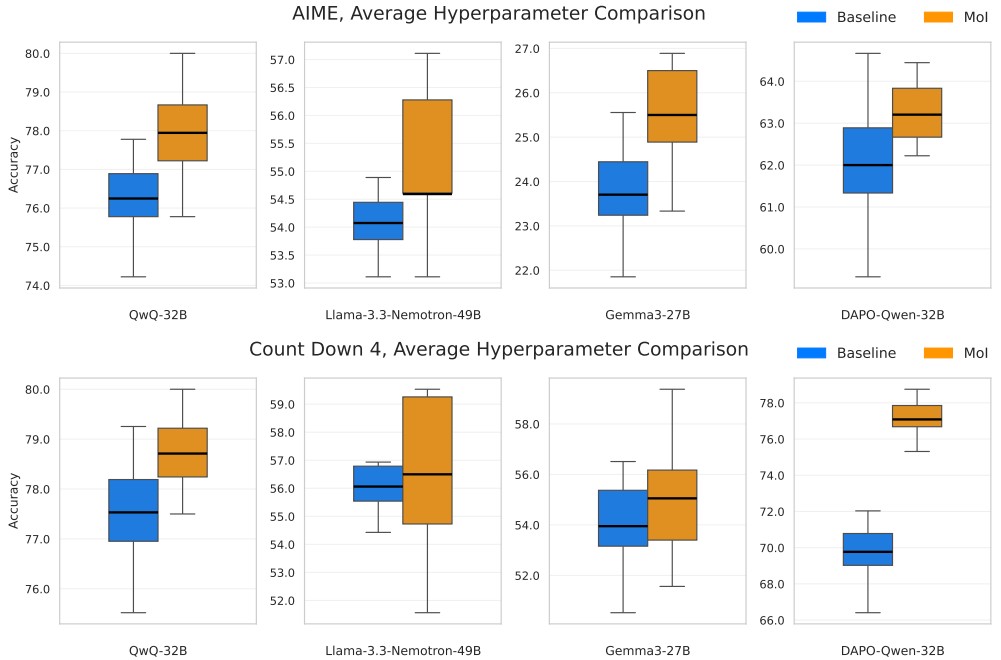

Figure A2: We show a comparison of distributions of evaluation results across all top-$p$ and temperature hyperparameters. The results indicate almost universal performance gain across average hyperparameter settings.

Table A1: Additional results on four benchmarks with four large language models. For every (method, model) we tune the decoding hyperparameters on AIME and CountDown 4, fix the best setting, and reuse it when assessing GPQA-Diamond and LiveCodeBench. Accuracy (%) is reported on AIME, Count Down 4, GPQA-Diamond, and pass@1 is used on LiveCodeBench. *Standard* uses conventional sampling, that is temperature–scaled nucleus sampling and MOI is our full approach.

| Model | Method | Input Info. | AIME | CountDown4 | GPQA-D | LiveCodeBench | Avg |
|---|---|---|---|---|---|---|---|
| **QwQ-32B** | Standard | Output Token | 76.89 | 78.04 | 58.08 | 76.48 | 72.37 |
| | MOI | Token + Dist. | 78.44 | 79.22 | 61.41 | 77.85 | 74.23 |
| | Gain vs. Standard | | **+1.55** | **+1.18** | **+3.33** | **+1.37** | **+1.86** |
| **Nemotron-Super-49B** | Standard | Output Token | 54.67 | 56.77 | 63.64 | 40.35 | 53.86 |
| | MOI | Token + Dist. | 57.11 | 59.22 | 64.95 | 40.90 | 55.55 |
| | Gain vs. Standard | | **+2.44** | **+2.45** | **+1.31** | **+0.55** | **+1.69** |
| **Gemma-3-27B** | Standard | Output Token | 24.44 | 55.21 | 45.45 | 32.37 | 39.37 |
| | MOI | Token + Dist. | 26.67 | 57.19 | 46.26 | 32.92 | 40.76 |
| | Gain vs. Standard | | **+2.23** | **+1.98** | **+0.81** | **+0.55** | **+1.39** |
| **DAPO-Qwen-32B** | Standard | Output Token | 64.67 | 70.31 | 42.93 | 56.91 | 58.70 |
| | MOI | Token + Dist. | 61.56 | 75.00 | 43.94 | 57.50 | 59.50 |
| | Gain vs. Standard | | **-3.11** | **+4.69** | **+1.01** | **+0.59** | **+0.80** |

decoding ($\beta$=1,2,4, same decoding configuration), together with the relative improvements brought by MOI ($\Delta$). As summarized in Table A2, MOI consistently outperforms the baseline for all models.

We also noticed MOI reduced repetition in the response. Particularly, we measured the 3-gram and 4-gram repetition rate in the response for MT-bench and found MoI reduces the repetition on 72.9% of responses, reducing the average 4-grams repetition rate from 4.27% to 3.13%. We also provide a side-by-side comparison on a coding task as below (QwQ-32B).

**With standard sampling (repetitive around HCA definition):**

*To address the problem of finding the highest common ancestor (HCA) of two nodes in a binary tree, we need to clarify the definition. Based on common terminology, the lowest common ancestor (LCA) is the deepest node shared by both paths from*

*the nodes to the root. However, since the problem specifies "not LCA," we assume the HCA refers to the shallowest common ancestor (closest to the root) [content continues...]*

**With mixture of inputs sampling (reduced repetitiveness):**

*To find the highest common ancestor (HCA) of two nodes in a binary tree, we can utilize a recursive approach similar to finding the Lowest Common Ancestor (LCA). The HCA is the deepest node that is an ancestor of both given nodes. Here's how to implement this: [content continues...]*

Table A2: Comparison of conventional sampling (Baseline) and MOI scores across six large language models. For each model, we report the baseline score, MOI score, and the relative improvement ($\Delta\%$) of MOI over the baseline.

| Model | Baseline Score | MOI Score | $\Delta\%$ (MOI vs Baseline) |
|---|---|---|---|
| QwQ-32B | 9.25 | 9.51 | **+2.81%** |
| Nemotron-Super-49B | 9.41 | 9.48 | **+0.74%** |
| Gemma-3-27B | 9.05 | 9.38 | **+3.65%** |
| DAPO-Qwen-32B | 8.96 | 9.46 | **+5.54%** |
| Llama-3.1-8B | 8.24 | 8.65 | **+4.98%** |
| Qwen-2.5-14B | 8.87 | 9.32 | **+5.07%** |

## D  Statistical Robustness of the Results

We conducted formal significance testing. Specifically, we applied McNemar's test to compare MOI against the baseline across all four benchmarks for each model. Table A3 reports the resulting p-values. These results validate that the observed gains are statistically significant in most cases.

To further address concerns about variance, we conducted an extensive 64-run evaluation on the AIME dataset across four models with the same configuration as in experiments of Table 1. The results in Table A4 confirm consistent improvements from MOI.

Table A3: McNemar's test $p$-values comparing model performance across four benchmarks. Lower values indicate statistically significant differences between baseline and MOI.

| Model | CountDown4 | AIME | GPQA-Diamond | LiveCodeBench |
|---|---|---|---|---|
| QwQ-32B | $< 0.001$ | $< 0.001$ | 0.003 | 0.04 |
| Nemotron-Super-49B | $< 0.001$ | $< 0.001$ | $< 0.001$ | 0.02 |
| Gemma-3-27B | $< 0.001$ | $< 0.001$ | 0.04 | $< 0.001$ |
| DAPO-Qwen-32B | $< 0.001$ | 0.8 | 0.02 | $< 0.001$ |

Table A4: Comparison of baseline and MOI performance across four large language models over 64 runs on AIME. Each result reports the mean ($\mu$), standard deviation ($\sigma$) and min-max score range. We observe consistent performance gain brought by MOI.

| Model | Baseline ($\mu \pm \sigma$) | Baseline Range | MoI ($\mu \pm \sigma$) | MoI Range | Gain |
|---|---|---|---|---|---|
| QwQ-32B | $76.02 \pm 2.51$ | 72.22–82.22 | $77.66 \pm 2.06$ | 73.33–82.22 | **+1.64** |
| Nemotron-Super-49B | $54.80 \pm 2.93$ | 48.89–61.11 | $55.03 \pm 3.54$ | 44.44–62.22 | **+0.23** |
| Gemma-3-27B | $21.80 \pm 2.34$ | 16.67–27.78 | $24.91 \pm 2.30$ | 20.00–30.00 | **+3.11** |
| DAPO-Qwen-32B | $61.16 \pm 2.50$ | 56.67–66.67 | $62.57 \pm 2.49$ | 57.78–68.89 | **+1.41** |

Table A5: Hyperparameter configuration by task. AIME and Count Down 4 use grid-search ranges; GPQA-Diamond and LiveCodeBench share a single universal setting.

| Hyperparameter | AIME | Count Down 4 | GPQA-D | LiveCodeBench |
|---|---|---|---|---|
| $\beta$ | $\{\frac{1}{4}, \frac{1}{2}, 1, 2, 4, 8\}$ | $\{\frac{1}{4}, \frac{1}{2}, 1, 2, 4, 8\}$ | 1 | 1 |
| Top-$p$ | $\{0.40, 0.60, 0.80, 0.95\}$ | $\{0.40, 0.60, 0.80, 0.95\}$ | 0.95 | 0.95 |
| Temperature $T$ | $\{0.6, 0.8, 1.0\}$ | $\{0.6, 0.8, 1.0\}$ | 0.6 | 0.6 |
| Max generation length | 32,768 | 8,192 | 16,384 | 16,384 |
| Chat template | Default Templates | Default Templates | No Chat Templates[†] | Default Templates |

[†]Except for Gemma-3-27B, the performance degradation is significant without chat template.

# E  Additional Setups

## E.1  Best-of-N Analysis Setup

To measure how quickly a limited tuning budget yields performance gains, we simulate a *best-of-N random search* for $N = 1, \ldots, 15$. For every model-task pair in AIME and Count Down 4 we start from the complete Cartesian grid of hyperparameters in Table 4. At each Monte-Carlo replicate we uniformly draw $N$ distinct values for a single target hyperparameter ($\beta$, top-$p$, or temperature) while keeping the other two at their default settings, retrieve the corresponding validation accuracies that were pre-computed during the main grid search, and record the improvement of the best sampled configuration over the initial draw. Repeating this procedure 256 times and averaging across the four LLMs produces the curves in Fig. 2.

## E.2  Random-Forest Regression Analysis Setup

Every completed grid-search run — defined by a specific model, task, and random seed—serves as one training example for a model-agnostic importance analysis. Each example is encoded by the triple ($\beta$, top-$p$, $T$) and labeled with its accuracy. We fit a `RandomForestRegressor` from SCIKIT-LEARN [44] with 100 trees that have unrestricted depth. Impurity-based Gini importances rank the hyperparameters as $\beta$ (0.41), top-$p$ (0.32), and temperature (0.27), as shown in Fig. 2.

## E.3  Setup for Case Study

Section 7.2 investigates linear prompt blending on five sentiment benchmarks with three 7B-sized LLMs: Llama3 8B [32], Mistral 7B [41] and Gemma 1.1 7B [42].

For each dataset, we sampled from GPT4o [45] and Claude [46] to curate a diverse pool of task prompts—96 for binary sentiment, 32 for six-class emotion, and 40 for financial sentiment—verifying that each prompt forms a syntactically valid query when concatenated with the input sentence. Letting $L_{\max}$ denote the length of the longest prompt in a pool, every prompt-embedding matrix $\mathbf{E} \in \mathbb{R}^{L \times d}$ is extended to length $L_{\max}$ via linear interpolation, and then combined by simple averaging across prompts to form the blended prompt.

During inference, we prepend this blended vector to each of the 10-shot demonstrations and feed the embeddings directly to the model. For the single-prompt baseline we calculate the average accuracy of the prompts as the expectation of randomly choosing any single prompt, whereas the blended prompt is evaluated once; consequently, Table 2 reflects an identical number of forward passes per model. Because all other factors remain fixed, any performance difference isolates the effect of the prompt representation itself.

# F  Hyperparameters

Table A5 lists the full search space for AIME and COUNT DOWN 4, along with the universal settings used for GPQA-DIAMOND and LIVECODEBENCH. All searches use five random seeds; reported results are seed-averaged.

# G   Implementation Details

We implement MOI on top of the vLLM framework [47], which supports efficient tensor parallelism. Mixing weights are computed from both the output token and the associated logits after each generation step. The resulting mixed inputs are cached and used as input for the subsequent decoding step.

For GPQA evaluations, we use the Language Model Evaluation Harness framework [48]. All models are evaluated using the default configuration with "thinking mode" enabled. The only exception is Gemma-3-27B, which requires an additional prompt to elicit multiple-choice outputs in the format of (A, B, C, D).

LiveCodeBench evaluations follow its official implementation [28], using the default setup and test suite corresponding to the period from May 2023 to May 2024. At the time of writing, generation templates were not officially available for Llama-3.3-Nemotron-Super-49B, Gemma-3-27B, and DAPO-Qwen-32B. We manually created a template for Llama-3.3-Nemotron-Super-49B based on its official documentation, including the required system prompt to activate its thinking mode. For Gemma-3-27B and DAPO-Qwen-32B, we adopt the `GenericBase` and `CodeQwenInstruct` templates, respectively.

