# OpenReview forum: "Mixture of Inputs: Text Generation Beyond Discrete Token Sampling"
_NeurIPS.cc/2025/Conference — NeurIPS 2025 poster_

### Official Review · Reviewer_Ev8i · 2025-06-28

**Clarity:** 2
**Significance:** 3
**Originality:** 3
**Rating:** 4
**Confidence:** 3

**Summary:**

This paper introduces a method that uses continuous tokens as input for large language models (LLMs), named Mixture of Inputs (MOI), to improve model performance. Specifically, it passes both the predicted distribution and the selected token information from the current step to the next step's input, employing a Dirichlet mixture model for mixing. Experimental results show that this method provides minor improvements in reasoning benchmarks across multiple models.

**Questions:**

See weaknesses

**Ethical Concerns:**

["NO or VERY MINOR ethics concerns only"]

**Final Justification:**

Although I am not fully persuaded regarding why this method brings improvements (and I still believe this should be a point that the paper needs to explore), the authors have provided fairly solid empirical results demonstrating the effectiveness of the approach. I will raise my score to 4.

**Limitations:**

The paper lacks deeper analysis on why using continuous embeddings improves performance as described in Weaknesses.

**Quality:**

3

**Strengths And Weaknesses:**

### Strengths

- The paper demonstrates that correctly mixing token embeddings can enhance LLM performance, whereas using direct linear interpolation of token embeddings leads to degradation.

### Weaknesses

- It is reasonable to build a one-hot token embedding and linearly interpolate it as model input. However, using a complex model like the Dirichlet mixture model lacks theoretical justification. Could a simpler interpolation achieve similar effects? For example, an interpolation formula such as $\alpha e_{\text{chosen}} + (1-\alpha) \sum w_i e_{i}$, where $\alpha$ is a hyperparameter, $e_{\text{chosen}}$ is the sampled token, and $w_i$ is the predicted distribution.

- The average improvement across four datasets is only around 2%, which raises two concerns:
  - Randomness in Small Sample Sizes: For datasets with fewer samples, such as AIME (with less than a hundred samples), single-generation randomness can introduce significant errors. The paper does not specify how many times AIME scores were averaged.
  - Lack of Significance Analysis: While the appendix (Appendix A) seems to provide distributions of multiple experimental results, it does not clearly state the number of experiments conducted. For all improvements listed in Table 1, variance and significance test results (e.g., t-tests) should be provided.

- The paper lacks deeper analysis on why using continuous embeddings improves performance. Models are trained using discrete one-hot tokens converted to embeddings as inputs, but this paper proposes a training-free approach by directly switching to continuous embeddings. Beyond accuracy, additional evidence supporting this improvement would be beneficial. Case studies would also provide valuable insights.

- Table 3 only reports changes in throughput. Does using this method affect the length of generated text?

---

> ### Author Rebuttal · Authors · 2025-07-31
>
> Thank you for your constructive feedback. We appreciate your comments and will address concerns regarding the Dirichlet model, intuition behind MoI, evaluation setup, and qualitative study in this rebuttal, with further elaborations to be included in the final paper.
>
> **Linear Interpolation and Dirichlet Posterior Mean** It is worth mentioning that the posterior mean of Dirichlet is a linear interpolation of the prior mean and the observation, and the mixing scheme proposed by the reviewer ($t \cdot e_{\mbox{chosen}} + (1 - t) \sum p_i e_i$, we replace $\alpha$ with $t$ since $\alpha$ has special mearnings in Dirichlet) can be viewed as a reparameterization of the posterior mean. Particularly, we assume $e_i$ to be the one-hot vector of $i$ for simplicity, then $t \cdot e_{\mbox{chosen}} + (1 - t) \sum p_i e_i$ is the posterior mean of the Dirichlet model $\alpha \to w \to e_{\mbox{chosen}}$, where $w \sim \mbox{Dir}(\frac{1-t}{t} p)$ and $e_{\mbox{chosen}} \sim \mbox{Multinomial}(w)$.
>
> Compared to the plain linear interpolation, the posterior mean yields the same results, but with a clearer interpretation. Specifically, the linear interpolation is viewed as the posterior mean, and the interpolation ratio is decided by the concentration factor. We believe this interpretation should be clean and understandable for the NLP community, where the Dirichlet prior is widely used, and we will add writing to make the connection to linear interpolation as simple and widely accessible as possible.
>
>
>
> **Length of generated text in throughput analysis** To better compare the throughput, we picked the benchmarks where the generation length is about the same. The median difference between MoI-generated text and baseline-generated text is 1.7%.
>
> **Significance of the improvements** We report the average over 5 independent runs for each decoding configuration (as stated in Section 5.4, line 213). In addition, we visualized performance distributions in Appendix A to give insight into variability across seeds.
>
> **Rational on why mixing continuous embeddings helped** Intuitively, continuous embedding is capable of capturing more information in the continuous state than the discrete token; there have been some recent theoretical analyses such as [1]. They effectively showed that continuous states can enable models to explore multiple generation paths at the same time, where discrete tokens need to collapse on one path at a time, hence can be more effective in finding a correct solution.
>
> We believe this continuous blending helps preserve distributional information that would otherwise be lost during hard sampling, enabling the model to produce more coherent and contextually consistent generations. Our empirical findings support this interpretation, particularly on tasks that benefit from structured reasoning (e.g., math and code).
>
> **Qualitative Analyses on Repetitiveness**
> We conducted additional qualitative analyses on MoI sampling outputs and noticed MoI reduced repetition in the response. Particularly, we measured the 3-gram and 4-gram repetition rate in the response for MT-bench and found MoI reduces the repetition on 72.9% of responses, reducing the average 4-grams repetition rate from 4.27% to 3.13%. We also provide a side-by-side comparison on a coding task as below (QwQ-32B).
>
> *With standard sampling (repetitive around HCA definition):*
> >To address the problem of finding the highest common ancestor (HCA) of two nodes in a binary tree, we need to clarify the definition. Based on common terminology, the lowest common ancestor (LCA) is the deepest node shared by both paths from the nodes to the root. However, since the problem specifies "not LCA," we assume the HCA refers to the shallowest common ancestor (closest to the root) [content continues...]
>
> *With mixture of inputs sampling (reduced repetitiveness):*
> >To find the highest common ancestor (HCA) of two nodes in a binary tree, we can utilize a recursive approach similar to finding the Lowest Common Ancestor (LCA). The HCA is the deepest node that is an ancestor of both given nodes. Here's how to implement this: [content continues...]
>
> **Additional results** We would like to highlight the additional experiments presented in our rebuttal:
>
>
>
> 1. Table A (in our reply to Reviewer Py1S) summarizes instruction following evaluation results on MT-Bench across 6 models, including 2 additional models that are smaller in sizes (i.e., Qwen-2.5-14B and LLaMA-3.1-8B), where MoI brings consistent performance gain.
>
> 2. Table B (in our reply to Reviewer Py1S) explores the robustness of Hyper-parameter tuning. It tunes hyperparameters on AIME/CountDown, and then directly transfers the resulting hyperparameters to GPQA and LiveCodeBench. MoI also leads to consistent performance gain under this setting.
>
> We hope our responses have adequately addressed your concerns and further highlighted the innovations and potential impact of our study. If you have any further questions or need additional information, please do not hesitate to ask.
>
> [1] Zhu, Hanlin, et al. "Reasoning by Superposition: A Theoretical Perspective on Chain of Continuous Thought." arXiv preprint arXiv:2505.12514 (2025).

---

> > ### Comment · Reviewer_Ev8i · 2025-08-02
> >
> > I still have two concerns that have not been fully addressed:
> >
> > * **Significance of Improvement**
> >
> >   * First, there is a clarity issue. Appendix A, especially Figure 4, lacks detailed descriptions of how the data was collected, including the number of sample points. Line 460 states only that "each method is run with its single best-performing hyperparameter configuration." However, distribution plots cannot be generated from just a single run.
> >   * For Table 1, reporting the average score over 5 runs is still insufficient to demonstrate statistical significance. Please include formal statistical tests, such as a t-test, to validate the improvements.
> >   * Please provide the number of test cases in each test set. On the AIME dataset, evaluating each sample only once typically leads to large score fluctuations (often exceeding 5 points) which is why most recent studies report metrics like AVERAGE@64. Yet, according to your figures, the variation appears to be around only 2 points. Please explain this discrepancy.
> >   * According to Table 13 in the Qwen3 Technical Report, QwQ-32B achieves 65.6 on GPQA-Diamond, but your Table 1 reports only 58.08. In addition, please specify the version used for the LiveCodeBench evaluation, as the scores differ significantly from those in the technical report.
> >
> > * **Lack of Deeper Insights**
> >
> >   * The authors state: "Intuitively, continuous embedding is capable of capturing more information in the continuous state than the discrete token; there have been some recent theoretical analyses such as [1]." However, [1]’s analysis is based on COCONUT, a method that uses continuous embeddings during training. In contrast, this work applies continuous embeddings only during inference using a pre-trained model, which introduces a train-test inconsistency. The paper does not offer sufficient discussion or justification for this mismatch.
> >   * Even though prior work has established the theoretical potential of continuous embeddings, this paper does not build on those insights or provide deeper analysis within its own proposed framework, such as under the Dirichlet mixture model. As a result, the overall contribution remains somewhat shallow and lacks conceptual depth.

---

> > > ### Author Response · Authors · 2025-08-04
> > >
> > > We appreciate your further inquiries. To answer your questions:
> > >
> > > - **On Experiments**
> > >
> > > 1. The data is collected in the following manner, we conduct grid search over hyperparameters for both baselines and MoI, for each hyperparameter configuration we repeat 5 runs. We plot the 5 runs of best configuration in Figure 4 (Best Scenarios Comparison), and the all the runs in Figure 5 (Avg Scenario Comparison). We will add this explanation in Appendix A.
> > >
> > > 2. As requested, we conducted formal significance testing. Specifically, we applied McNemar’s test to compare MoI against the baseline across all four benchmarks for each model. The table below reports the resulting p-values:
> > >
> > > **Table C: McNemar’s test**
> > > | Model                  | CountDown4 |    AIME | GPQA-Diamond | LiveCodeBench |
> > > | ---------------------- | ---------: | ------: | -----------: | ------------: |
> > > | **QwQ-32B**            |    < 0.001 | < 0.001 |        0.003 |          0.04 |
> > > | **Nemotron-Super-49B** |    < 0.001 | < 0.001 |      < 0.001 |          0.02 |
> > > | **Gemma-3-27B**        |    < 0.001 | < 0.001 |         0.04 |       < 0.001 |
> > > | **DAPO-Qwen-32B**      |    < 0.001 |     0.8 |         0.02 |       < 0.001 |
> > >
> > > These results validate that the observed gains are statistically significant in most cases.
> > >
> > > 3. To further address concerns about variance, we conducted an extensive 64-run evaluation on the AIME dataset across four models with same configuration as Table 1. The results below confirm consistent improvements from MoI:
> > >
> > > **Table D: AIME Evaluation with 64 runs**
> > > | Model                  | Baseline, Mean ± STD | Baseline Range | MoI Mean ± STD | MoI Range |      Gain |
> > > | ---------------------- | --------------------: | ----------------: | ---------------: | -----------: | --------: |
> > > | **QwQ-32B**            |          76.02 ± 2.51 |       72.22–82.22 |     77.66 ± 2.06 |  73.33–82.22 | **+1.64** |
> > > | **Nemotron-Super-49B** |          54.80 ± 2.93 |       48.89–61.11 |     55.03 ± 3.54 |  44.44–62.22 | **+0.23** |
> > > | **Gemma-3-27B**        |          21.80 ± 2.34 |       16.67–27.78 |     24.91 ± 2.30 |  20.00–30.00 | **+3.11** |
> > > | **DAPO-Qwen-32B**      |          61.16 ± 2.50 |       56.67–66.67 |     62.57 ± 2.49 |  57.78–68.89 | **+1.41** |
> > >
> > >
> > > 4. We used the default evaluation script from the lm-evaluation-harness without custom prompt formatting or chat templates (as stated in Table 4). Our results (e.g., 58.08 for QwQ-32B) align with independent studies, such as the ACL 2025 paper [2] (58.05%) and the Artificial Analysis report [3] (59%). We have included the information of LiveCodeBench and other evaluation benchmarks in Appendix D, we used the version from May 2023 to May 2024. We will also explicitly include the number of examples per benchmark in the revision.
> > >
> > > - **On Deeper Insights and Conceptual Contributions**
> > >
> > > Reference [1] provides theoretical analysis demonstrating that information can exist in superposition states within Transformers, enabling more effective reasoning. **Our work extends this insight by showing that interpolating within the convex hull of token representations achieves similar superposition effects without requiring expensive retraining.**
> > >
> > > This connection is grounded in the observation that modern language models already operate on continuous representations internally (e.g., in attention and MLP layers). By forming convex combinations of pretrained input embeddings, we directly exploit this geometric structure.
> > >
> > > We demonstrate that this principle works at multiple levels:
> > > 1. Token-level (main method): Individual tokens can be represented as continuous mixtures, enabling fine-grained semantic interpolation
> > > 2. Prompt-level (Section 7.2): Our case study reveals latent ensembling effects when interpolating across multiple variable-length prompts.
> > >
> > > While we do not introduce a new training paradigm, our findings offer a new test-time perspective on continuous reasoning capabilities in LLMs. This highlights a complementary research direction to existing training-based approaches.
> > >
> > > Thank you for your thoughtful follow-up. We appreciate the opportunity to clarify and expand upon the concerns you raised.
> > >
> > > [2] Zeng, Zhiyuan, et al. "Revisiting the Test-Time Scaling of o1-like Models: Do they Truly Possess Test-Time Scaling Capabilities?." arXiv preprint arXiv:2502.12215 (2025).
> > >
> > > [3] QWQ-32B - Intelligence, Performance Analysis. Artificial Analysis. (2025)

---

> > > > ### Comment · Reviewer_Ev8i · 2025-08-04
> > > >
> > > > Thanks for the responses. Although I am not fully persuaded regarding why this method brings improvements (and I still believe this should be a point that the paper needs to explore), the authors have provided fairly solid empirical results demonstrating the effectiveness of the approach. I will raise my score to 4.

---

> > > > > ### Author Response · Authors · 2025-08-04
> > > > >
> > > > > We appreciate your decision to raise your score and again thank you for your constructive feedback.

---

### Official Review · Reviewer_gMqW · 2025-07-01

**Clarity:** 3
**Significance:** 4
**Originality:** 4
**Rating:** 5
**Confidence:** 4

**Summary:**

The paper proposes Mixture of Inputs, where instead of passing in a one-hot encoding the sampled token, they instead pass in a linear mixture of the sampled one-hot and the predicted next token distribution. The method is motivated by the intuition that models might perform better if they can explore multiple reasoning paths instead of being forced into a single path. The authors evaluate MoI on a wide range of models and tasks, finding that it consistently improves performance.

**Questions:**

(1) For the direct mixture baseline, to make things fair with the standard sampling baseline, I would be curious about how much things improve if one also does grid search on nucleus top-p and temperature.

(2) Why do you do grid search on aime and countdown, but not on gpqa and livecodebench?

(3) How important is it to have entropy-dependent mixture weights? How well would the method perform if one simply did a linear mixture with some fixed weight? And what if we normalize it to [0,2] instead of [0,1]? Why was this particular normalization chosen?

(4) The Dirichlet model makes sense in that the posterior mean just ends up being a linear combination, but I don't really understand the motivation or intuition behind this probabilistic model. For example, why do we not see w, when it is observed in practice? What are we uncertain over?

**Ethical Concerns:**

["NO or VERY MINOR ethics concerns only"]

**Final Justification:**

The new experiments look good and most of my questions were answered. I still find the Dirichlet model a bit unnatural but the authors' explanation was helpful. I will keep my score the same (5 - Accept).

**Limitations:**

yes

**Quality:**

4

**Strengths And Weaknesses:**

Strengths:

(1) The method is interesting, simple, and requires no additional training. It also has strong results across a wide range of models and datasets.

(2) The paper has thorough analysis and ablations, with interesting intuition about the mixture weight beta across different tasks.

Weaknesses: no major weaknesses; some minor ones are discussed in the questions section.

---

> ### Author Rebuttal · Authors · 2025-07-31
>
> Thank you for your constructive feedback. We appreciate your comments and will address concerns regarding evaluation setup, the role of entropy in MoI, and the intuition of the Dirichlet model.
>
> **Response to Question 1 on baseline setup** Both the vanilla sampling and the direct mixture baseline were also tuned over grid search for top-p and temperature. We will add an explanation in the paper to clarify this.
>
> **Response to Question 2 on evaluation setup** We deliberately choose not to do grid search on GPQA and LiveCodeBench to demonstrate that MoI does not rely heavily on hyperparameter tuning.
>
> **Response to Question 3.a on the importance of entropy** In our early phase experiments, we had explored replacing entropy with a hyper-parameter, which causes QwQ-32B to drop 7.07% on GPQA-Diamond. After some analyses, we found it is necessary to balance the output distribution and one-hot output token, for which we found entropy to be a natural choice.
>
> **Response to Question 3.b on the design of normalization** We chose the [0,1] range to ensure all token weights remain non-negative and interpretable as probabilities. Extending the range or allowing negative weights is possible, though we suspect this may complicate optimization and introduce instability. Exploring more flexible formulations would be an interesting direction for future work.
>
> **Response to Question 4.a on the intuition behind the Dirichlet model** At the input, the model has both the predicted distribution and the sampled discrete token. To capture the semantic meaning of both, we need to “adapt” the predicted distribution so that the resulting distribution incorporates the information of the sampled discrete token. To this end, we find it is a natural idea to use the Dirichlet model. As widely used in topic modeling (LDA), the Dirichlet distribution is the natural prior over multinomial probabilities and gives closed-form posteriors (due to conjugacy).
>
> **Response to Question 4.b on the predicted distribution p** The predicted distribution p represents the semantics of model outputs *before* we commit to a sampling trajectory. Using p directly as w would ignore the valuable signal from the sampling decision. In our experiments, we refer to this approach as *Direct Mixture* and find MoI to outperform this baseline consistently (as in Table 1 of the submission).
>
>  **Response to Question 4.c on the uncertainty** Although we observed both (1) the model's predicted distribution, and (2) the sampled token, it is still an open question to find a distribution that incorporates the semantics of both the distribution and the token. Here, we assume such an underlying distribution exists, and propose to use the Dirichlet method to estimate it.
>
> **Additional results** We would like to highlight the additional experiments presented in our rebuttal:
>
>
> 1. Table A (in our reply to Reviewer Py1S) summarizes instruction following evaluation results on MT-Bench across 6 models, including 2 additional models that are smaller in size (i.e., Qwen-2.5-14B and LLaMA-3.1-8B), where MoI brings consistent performance gain.
>
> 2. Table B (in our reply to Reviewer Py1S) explores the robustness of Hyperparameter tuning. It tunes hyperparameters on AIME/CountDown, and then directly transfers the resulting hyperparameters to GPQA and LiveCodeBench. MoI also leads to consistent performance gain under this setting.
>
> 3. In the *Qualitative Analyses on Repetitiveness* part of our reply to Reviewer Py1S, we observe that MoI helps to reduce the repetitiveness in the model generations.
>
> We hope our responses have adequately addressed your concerns and further highlighted the innovations and potential impact of our study. If you have any further questions or need additional information, please do not hesitate to ask.

---

> > ### Comment · Reviewer_gMqW · 2025-08-01
> >
> > Thanks for the response! The new experiments look good and most of my questions were answered. I still find the Dirichlet model a bit unnatural but your explanation was helpful. I will keep my score the same (5 - Accept).

---

> > > ### Author Response · Authors · 2025-08-01
> > >
> > > We thank you for your timely response. We're glad our clarifications were helpful. We will incorporate our new results and explanations in the revision.

---

### Official Review · Reviewer_Py1S · 2025-07-03

**Clarity:** 3
**Significance:** 3
**Originality:** 3
**Rating:** 5
**Confidence:** 4

**Summary:**

This paper proposes a novel generative sampling method. It utilizes Bayesian estimation to combine the predicted probability distribution from the previous step with the actually sampled token, creating a new input representation. This approach requires no additional training and has minimal computational overhead. Experimental results show that MOI significantly improves the performance of various LLMs on multiple tasks, including mathematical reasoning, code generation, and PhD-level question answering.

**Questions:**

1. Why were there no experiments conducted for the case where \beta = 0 ?

2. In terms of model selection, why was the Qwen series of models not included in the evaluation, while QwQ were?

3. In Equation 5, what would happen if we were to remove H and only use a hyperparameter to control the proportion between p_t and y_t?

**Ethical Concerns:**

["NO or VERY MINOR ethics concerns only"]

**Final Justification:**

The additional experiments have addressed all my concerns.

**Limitations:**

yes

**Quality:**

3

**Strengths And Weaknesses:**

**Strengths:**

1.  The method is simple and plug-and-play, and it demonstrates strong performance across a wide range of tasks.

2.  The experimental analysis is thorough, and the method is well-motivated with a clear theoretical foundation.

**Weaknesses:**

1.  Although the authors have already mentioned this in the limitations section, I still have questions regarding the method's generality. The paper lacks a systematic experimental comparison on open-domain dialogue. In fact, numerous evaluation benchmarks are available, such as MT-Bench, AlpacaEval, and the LMSYS Chatbot Arena, which can be used to assess human preferences. For the completeness of the paper, conducting such experiments seems essential.

2.  The sensitivity to the hyperparameter \beta is indeed a concern. It would be beneficial if the authors could provide suggested values for \beta for a broader range of general-purpose tasks.

3.  The paper lacks qualitative analysis or case studies. Observing the generation trajectories of specific examples and providing a comparative analysis would be an interesting experiment. This would offer valuable insights into how the method works in practice.

---

> ### Author Rebuttal · Authors · 2025-07-31
>
> We thank you for the constructive feedback.  We appreciate your comments and will address concerns regarding instruction-following capacity, hyper-parameter selection, and qualitative study in this rebuttal, with further elaborations to be included in the final paper.
>
> **Additional results on instruction-following benchmarks and more models** We conducted additional experiments on MT-Bench, using the four models from our original paper, and adding two smaller models, Llama-3.1-8B and Qwen-2.5-14B. Below, we report the average performance for standard decoding (temperature=0.6, top-p=0.95) and our Mixture of Inputs (MoI) decoding (beta={1,2,4}, same decoding config). together with the relative improvements brought by MoI (Δ). As summarized in Table A, MoI consistently outperforms the baseline for all models.
>
> **Table A: MT-Bench Scores**
> | Model            | Baseline Score | MoI Score | Δ % (MoI vs Baseline) |
> |------------------|:--------------:|:---------:|:---------------------:|
> | DAPO‑Qwen‑32B    | 8.96 | 9.46 | **+5.54 %** |
> | QwQ‑32B          | 9.25 | 9.51 | **+2.81 %** |
> | Nemotron‑49B     | 9.41 | 9.48 | **+0.74 %** |
> | Gemma‑27B        | 9.05 | 9.38 | **+3.65 %** |
> | Llama‑3.1‑8B     | 8.24 | 8.65 | **+4.98 %** |
> | Qwen‑2.5‑14B     | 8.87 | 9.32 | **+5.07 %** |
>
>
> **Hyperparameter Robustness** In our experiments, we observed that it generally improves the performance across tasks by setting beta to less aggressive values (0.5-2), while tuning the hyperparameter has the potential to further push the performance for each task. To demonstrate the robustness of the hyperparameter, we conduct a hyperparameter transfer experiment as below.
>
> In particular, we first choose the hyperparameter (beta/temperature/top-p) based on the performance on AIME & CountDown-4, and test it on GPQA-Diamond and LiveCodeBench. As summarized in Table B, MoI consistently outperforms the baseline for all models.
>
> **Table B: Hyperparameter Robustness (hyperparameters used in this experiment is picked based on their AIME & CountDown-4 performance)**
> | Model                  | Method   | Input Info    |    GPQA-D |       LCB |   Avg |
> | ---------------------- | -------- | ------------- | --------: | --------: | ----: |
> | **QwQ-32B**            | Standard | Output Token  |     58.08 |     76.48 | 67.28 |
> |                        | **MoI**  | Token + Dist. | **61.41** | **77.85** | 69.63 |
> |                        | Gain     | —             |     +3.33 |     +1.37 | +2.35 |
> | **Nemotron-Super-49B** | Standard | Output Token  |     63.64 |     40.35 | 52.00 |
> |                        | **MoI**  | Token + Dist. | **64.95** | **40.90** | 52.92 |
> |                        | Gain     | —             |     +1.31 |     +0.55 | +0.92 |
> | **Gemma-3-27B**        | Standard | Output Token  |     45.45 |     32.37 | 38.91 |
> |                        | **MoI**  | Token + Dist. | **46.26** | **32.92** | 39.59 |
> |                        | Gain     | —             |     +0.81 |     +0.55 | +0.68 |
> | **DAPO-Qwen-32B**      | Standard | Output Token  |     42.93 |     56.91 | 49.92 |
> |                        | **MoI**  | Token + Dist. | **43.94** | **57.50** | 50.72 |
> |                        | Gain     | —             |     +1.01 |     +0.59 | +0.80 |
>
>
>
> **Qualitative Analyses on Repetitiveness**
> We conducted additional qualitative analyses on MoI sampling outputs and noticed MoI reduced repetition in the response. Particularly, we measured the 3-gram and 4-gram repetition rate in the response for MT-bench and found MoI reduces the repetition on 72.9% of responses, reducing the average 4-gram repetition rate from 4.27% to 3.13%. We also provide a side-by-side comparison on a coding task as below (QwQ-32B).
>
> *With standard sampling (repetitive around HCA definition):*
> >To address the problem of finding the highest common ancestor (HCA) of two nodes in a binary tree, we need to clarify the definition. Based on common terminology, the lowest common ancestor (LCA) is the deepest node shared by both paths from the nodes to the root. However, since the problem specifies "not LCA," we assume the HCA refers to the shallowest common ancestor (closest to the root) [continues...]
>
> *With mixture of inputs sampling (reduced repetitiveness):*
> >To find the highest common ancestor (HCA) of two nodes in a binary tree, we can utilize a recursive approach similar to finding the Lowest Common Ancestor (LCA). The HCA is the deepest node that is an ancestor of both given nodes. Here's how to implement this: [continues...]
>
> **Response to Question 1: what about Beta=0?** We think beta=0 would be similar to the case of the direct mixture baseline. Beta=0 would fully emphasize the predicted distribution and ignore the sampled token. We found this often leads to less grounded and unstable generations, so we focused on more practical beta values that blend both signals.
>
> **Response to Question 2:  why not Qwen?** QwQ-32B is the leading open-source mode at its 32B scale from the Qwen family at the time of our submission. In Table A, we included a Qwen-2.5-14B model in our MT-bench experiments and found MoI consistently brings performance to this model.
>
> **Response to Question 3: what about removing H?** In our early phase experiments, we had explored replacing H with a hyperparameter, which causes QwQ-32B to drop 7.07% on GPQA-Diamond. After some analyses, we found it is necessary to balance the output distribution and one-hot output token dynamically, for which we found entropy to be a natural choice.
>
> We hope our responses have adequately addressed your concerns and further highlighted the innovations and potential impact of our study. If you have any further questions or need additional information, please do not hesitate to ask.
>
> [1] Zhu, Rui-Jie, et al. "A survey on latent reasoning." arXiv preprint arXiv:2507.06203 (2025).

---

> > ### Comment · Reviewer_Py1S · 2025-08-04
> >
> > Thank you for your response. The additional experiments have addressed my concerns, and I will raise my score accordingly. I hope the authors will include these experiments in the revised version of the manuscript.

---

> > > ### Author Response · Authors · 2025-08-04
> > >
> > > Thanks again for your constructive feedback. We will include these discussions and additional results in the revision.

---

### Official Review · Reviewer_3KDf · 2025-07-04

**Clarity:** 4
**Significance:** 3
**Originality:** 4
**Rating:** 5
**Confidence:** 4

**Summary:**

This paper proposes Mixture of Inputs (MoI), a method that blends the discrete one-hot vector and the continuous next-token probability distribution in autoregressive language modeling. The authors uses a dirichlet distribution to interpolate between the mixture between the predicted one-hot distribution and the full probability distribution. MoI allows the model to maintain richer internal representation compared to standard autoregressive modeling. Experiments spans multiple downstream tasks and models show the effectives of the method compared to standard decoding.

**Questions:**

1. Would the authors be able to show a few generated examples with both standard decoding and the proposes method side by side to see where does the proposed method actually improve?

2. Can the authors explain how to integrate their method to vLLM?

Formatting issue: Figure 3 on page 9 is a bit too large, consider making it smaller.

**Ethical Concerns:**

["NO or VERY MINOR ethics concerns only"]

**Limitations:**

Yes.

**Paper Formatting Concerns:**

N/A.

**Quality:**

4

**Strengths And Weaknesses:**

Strengths:

- The method is novel and interesting: too much information is discarded by just feeding the predicted token back to the model.
- Experiments are comprehensive, covering many downstream tasks and model families. The performance improvement is quite large given that this method does not incur too much overhead to standard decoding.
- The paper is clearly written, with details and hyperparameters that enhances the reproducibility of the work.

Weaknesses:

- The experiments are mostly conducted on verifiable benchmarks. However, a lot of the LM use cases are chatbots. It would make the paper much more solid if the author can conduct experiments on an instruction following benchmark (e.g. WildBench), or show a few (not cherry picked) examples of the generations to check the quality.

- The significance of the work is questionable: my understanding is that this method requires a good estimate of the next token distribution (log probs). But modern inference engines (e.g. vLLM, sglang) uses a lower precision, and that the full probability distritbution from these inference engines might not be as accurate. If this method is incompatible with inference engines, I don't know how much would the community adopt this method.

---

> ### Author Rebuttal · Authors · 2025-07-31
>
> Thank you for your constructive feedback. We appreciate your comments and will address concerns regarding inference engine compatibility, instruction-following capacity, qualitative study in this rebuttal, with further elaborations to be included in the final paper.
>
> **Modern inference engine compatibility**  We agree modern inference engines play an important role in LLM serving, and accordingly build our implementation on top of vLLM, despite it requiring more engineering efforts. In fact, all experiments in our submission are conducted on our vLLM-based implementation, and the baseline method in the throughput analyses is the original vLLM. As to the precision issue, the vLLM V1 engine’s sampler casts the softmax logits to fp32 for sampling (as in L47 of the `sampler.py` in the vLLM repo), and we followed the same strategy and didn’t notice any significant issues.
>
> **Additional results on instruction-following benchmarks and more models** We conducted additional experiments on MT-Bench, using the four models from our original paper, and adding two smaller models, Llama-3.1-8B and Qwen-2.5-14B. Below, we report the average performance  between standard decoding (temperature=0.6, top-p=0.95) and our Mixture of Inputs (MoI) decoding (beta={1,2,4}, same decoding config), together with the relative improvements brought by MoI (Δ). As summarized in Table A, MoI consistently outperforms the baseline for all models.
>
> **Table A: MT-Bench Scores**
> | Model            | Baseline Score | MoI Score | Δ % (MoI vs Baseline) |
> |------------------|:--------------:|:---------:|:---------------------:|
> | QwQ‑32B          | 9.25 | 9.51 | **+2.81 %** |
> | Nemotron‑49B     | 9.41 | 9.48 | **+0.74 %** |
> | Gemma‑27B        | 9.05 | 9.38 | **+3.65 %** |
> | DAPO‑Qwen‑32B    | 8.96 | 9.46 | **+5.54 %** |
> | Llama‑3.1‑8B     | 8.24 | 8.65 | **+4.98 %** |
> | Qwen‑2.5‑14B     | 8.87 | 9.32 | **+5.07 %** |
>
>
> **Qualitative Evaluation on Repetitiveness** We conducted additional qualitative analyses on MoI sampling outputs and noticed MoI reduced repetition in the response. Particularly, we measured the 3-gram and 4-gram repetition rate in the response for MT-bench and found MoI reduces the repetition on 72.9% of responses, reducing the average 4-grams repetition rate from 4.27% to 3.13%. We also provide a side-by-side comparison on a coding task as below (QwQ-32B).
>
> *With standard sampling (repetitive around HCA definition):*
> >To address the problem of finding the highest common ancestor (HCA) of two nodes in a binary tree, we need to clarify the definition. Based on common terminology, the lowest common ancestor (LCA) is the deepest node shared by both paths from the nodes to the root. However, since the problem specifies "not LCA," we assume the HCA refers to the shallowest common ancestor (closest to the root) [content continues...]
>
> *With mixture of inputs sampling (reduced repetitiveness):*
> >To find the highest common ancestor (HCA) of two nodes in a binary tree, we can utilize a recursive approach similar to finding the Lowest Common Ancestor (LCA). The HCA is the deepest node that is an ancestor of both given nodes. Here's how to implement this: [content continues...]
>
> We hope our responses have adequately addressed your concerns and further highlighted the innovations and potential impact of our study. If you have any further questions or need additional information, please do not hesitate to ask.

---

### Decision · Program_Chairs · 2025-09-17

**Decision:**

Accept (poster)

**Comment:**

The paper presents a new method that combines the token distribution and the sampled token for better autoregressive generation. The reviewers in general value the proposed idea and find the empirical evaluation to be fairly solid. They raised concerns such as its compatibility with modern inference engines, the performance on the instruction following benchmarks, and significance and insights on the improvement etc.

The author provided additional results and clarification on these concerns, and the rebuttal convinced the negative reviewer **Ev8i** towards positive. The AC agrees with the reviewers and values the proposed model and its strong empirical validation, thus recommending acceptance of the paper. The final version should, as suggested by the reviewers, include the results in the rebuttal and provide well-articulated arguments on the motivation of using the Dirichlet model as well as insights on the model improvement.